

# The importance of moist thermodynamics on neutral buoyancy height for plumes from anthropogenic sources

Sepehr Fathi[1], Paul Makar[1], Wanmin Gong[1], Junhua Zhang[1], Katherine Hayden[1], and Mark Gordon[2]

[1]Air Quality Research Division, Environment and Climate Change Canada, Toronto, Canada
[2]Earth and Space Science, York University, Toronto, Canada

**Correspondence:** Sepehr Fathi (sepehr.fathi@ec.gc.ca)

**Abstract.**

Plume rise plays a critical role in dispersing pollutants emitted from tall stacks, dictating the height reached by buoyant plumes and their subsequent downwind dispersion. Commonly, plume rise is assumed to be governed by atmospheric stability, the exit momentum and temperature of the effluent released from large stacks. However, an under-recognized influence on plume rise is the effects of entrained and/or co-emitted water, which can change the plume height due to exchange of latent heat associated with phase changes of within-plume water. While many of the stack sources achieve high temperatures of the emitted effluent via combustion, the impact of combustion-generated water on plume rise is often overlooked in large-scale air-quality models. As the rising water condenses or evaporates, it releases or absorbs latent heat, influencing the height reached by the plumes. Our study investigates the effects of latent heat exchange by combustion-generated and entrained water on plume rise. We introduce a novel approach that integrates moist thermodynamics into an empirical parameterization for plume rise, resulting in the development of **PRISM** (Plume-Rise-Iterative-Stratified-Moist). Long-term (6-month duration) simulations using PRISM exhibit a difference of up to $\pm 100\%$ in surface concentrations of emitted pollutants near industrial sources compared to previous predictions, emphasizing the substantial influence of moist thermodynamics on plume rise. Our results show up to 50% improvement in model simulated plume height, through evaluation against aircraft observations over the Canadian Oil Sands. This study pioneers a plume rise sub-grid parameterization integrating moist thermodynamics in iterative calculation of neutral buoyancy height for plumes emitted from industrial stacks, thereby advancing our understanding of plume behaviour and enhancing the accuracy of air-quality modelling. These advancements can potentially contribute to more effective pollution control strategies.

## 1 Introduction

Effluents emitted from industrial and urban sources (e.g., stacks) are often much warmer than the surrounding air, and therefore buoyant. If the source of heat for the effluent is the combustion of hydrocarbons, in which water is a by-product of combustion, then the water content of the rising plume may be greater than that of the surrounding atmosphere. The emitted effluents rise to higher altitudes than the original release height due to exit momentum and buoyancy, while simultaneously the water vapour content condenses (as plumes expand and cool) forming the visible (cloud like) plumes that can be observed rising from



chimney stacks and other sources (e.g., Sturman and Zawar-Reza, 2011). The buoyant rise due to the effluent's exit velocity
and temperature upon emissions is captured within standard algorithms of plume rise (e.g., Briggs, 1984). However, the effects
of latent heat exchange due to water condensation into droplets and evaporation of these droplets for plumes emitted from
industrial stacks has not been implemented as a controlling variable in plume rise sub-grid parameterization for air quality
models. Through three-dimensional numerical modelling of the governing processes (e.g., mass and energy balance), Gangoiti

et al. (1997) has shown the impact of latent heat exchange on plume buoyancy and atmospheric dispersion for plumes from tall
stacks. However, computational costs prevent the use of explicit numerical modelling of plume trajectory for regional large-
scale air quality models with grid sizes of a few kilometres and domain sizes of thousands of kilometres, where plumes from
thousands of simultaneously emitting sources may be simulated. For these regional chemical transport models (e.g., CMAQ,
GEM-MACH), plume rise is usually determined using some form of sub-grid parameterization embedded within the host 3D

model (e.g., Briggs, 1984). We note that latent heat effects have been previously taken into account in plume rise parameteriza-
tion for vegetation (wild) fires (e.g., Freitas et al., 2007; Chen et al., 2019). However, sub-grid parameterizations in large-scale
air quality (chemical transport) models commonly don't incorporate moist thermodynamics in estimating plume rise from high
temperature industrial stacks. The transport of the emitted pollutants are governed by meteorological conditions and atmo-
spheric flow regimes (wind speed and direction) at the effective release height. Therefore, to reliably predict the range/extent

of the atmospheric dispersion of the emitted pollutants, accurate plume rise parameterization is essential and has important im-
plications for air quality predictions. For instance, determining the final plume rise (sometimes referred to as the effective stack
height) is a requirement for the estimation of the maximum surface concentration at distances downwind of the emission source.
Calculating the final rise with acceptable certainty is more difficult for unstable (convective) conditions where turbulence is
the main rise limiting factor (the rise may never actually terminate), compared to stable-atmosphere conditions with low winds

(Briggs, 1984). Since the 1960s, much research work have been dedicated to plume rise parameterization through dimensional
analysis where empirical parameters are determined from laboratory measurements and field observations (Hoult et al., 1969).
Many air-quality models (e.g., Im et al., 2015; Byun and Ching., 1999; Holmes and Morawska, 2006) use a variation of empiri-
cal formulations developed by G. A. Briggs during late 1960s to early 1980s (e.g., Briggs, 1965, 1969, 1975, 1984), such as the
Community Multiscale Air Quality (CMAQ;  Byun and Schere, 2006) and Global Environmental Multiscale – Modelling Air-

quality and Chemistry (GEM-MACH;  Moran et al., 2010) models. Briggs' (1984) empirical formulations parameterize plume
rise based on estimates of meteorological conditions (e.g., stability) at stack location/height, source information (e.g., stack
flow rate, temperature), estimated entrainment rates, and data on observed plume heights. Briggs' formulations (and most other
plume rise parameterizations) assume uniform meteorological conditions (e.g., temperature, wind speed) over the vertical span
of the plume, either taken at the stack top or averaged over the atmospheric layers between bottom and top of the plume. Such

simplifications, when applied to cases where the atmospheric vertical structure is complex, can lead to large errors in plume
final rise estimation. While commonly employed, subsequent evaluations of such parameterizations have shown over-/under-
predictions by over 50% against observed plume heights (e.g., Hamilton, 1967; England et al., 1976; Rittmann, 1982; Webster
and Thomson, 2002). Gordon et al. (2018) conducted extensive evaluations of plume rise prediction using Briggs (1984) for-
mulation driven by ambient observations, against aircraft $SO_2$ measurements over Canadian Oil Sands (OS) during the JOSM



2013 campaign (ECCC, 2018). They found Briggs (1984) plume-rise algorithm to significantly under-predict the observed $SO_2$ plume heights, with more than 50% of the predicted plume heights less than half that of observed heights for plumes from large $SO_2$ emitting OS sources. Results by Gordon et al. (2018) also included a subset of cases (less than 12%) with over-predicted plume heights, where plume height predictions by Briggs (1984) algorithm were more than twice the observed $SO_2$ plume heights. These discrepancies were partially attributed to potential presence of spatial heterogeneity in meteorological data used

to drive the plume rise algorithm (input data were not co-located with the emission stacks). The impact of spatial heterogeneity was confirmed by Akingunola et al. (2018) through high resolution meteorological model simulations for the same locations and time periods. Akingunola et al. (2018) demonstrated, using model-generated meteorological conditions at stack locations and calculations of residual plume buoyancy at successive levels above the inversion layer height, that incorporation of these factors into a plume rise model can significantly improve plume rise predictions, with 70% of predictions falling within a factor

of 2 of the observed plume heights.

   Utilizing more accurate source emissions information (e.g., Continuous Emission Monitoring system - CEMS) and source specific meteorology can improve the confidence in initial/input information for plume rise parameterization, while a layered approach can better resolve plume buoyancy in cases of more complex atmospheric conditions. However, efforts in improving plume rise parameterization (for large-scale air quality models) have largely ignored the potential importance of (within-

plume) water thermodynamic effects. Plume buoyancy is commonly determined in terms of initial stack exit temperature and buoyancy flux reduction as the plume rises along with estimates of the ambient temperature gradient (i.e., the height at which the plume comes to rest having the same temperature as the ambient atmosphere). However, as we show in the following work, release and/or uptake of the latent heat associated with phase changes of water can potentially alter plume buoyancy enough to impact the plume rise significantly. In this work we introduce a new plume rise algorithm that performs plume

buoyancy calculations at all vertical levels above the stack top (as opposed to Akingunola et al. (2018) where plume residual buoyancy calculations are done only above the inversion layer height), which also accounts for the effect of latent heat exchange associated with phase changes of within-plume water content. This algorithm expands on relevant concepts from Briggs (1984) and Akingunola et al. (2018), while including estimates of water emissions (due to combustion) from stack sources in a new plume rise parameterization. Following comparisons of predicted plume heights using an observation-driven model against

observed heights, we implemented the new parameterization within the GEM-MACH air-quality model (Moran et al., 2010; Makar et al., 2021) and conducted a series of retrospective air-quality model simulations for the Athabasca Oil Sands (OS) region. We considered a simulation period which overlaps with that of a 2018 aircraft measurement campaign over OS as part of the Canada-Alberta Oil Sands Monitoring program (OSM; ECCC, 2018). We conducted sensitivity analyses on the plume rise parameterization, and evaluated model performance against surface monitoring data and aircraft measurements.



## 2 Methods

### 2.1 PRISM (Plume-Rise-Iterative-Stratified-Moist): The New Algorithm for Plume Rise Parameterization

We developed a plume rise prediction algorithm based on effluent buoyancy-flux reduction while accounting for thermodynamic effects associated with latent heat release/uptake as described below. The stack parameters such as stack radius, exit momentum and temperature are translated into effluent initial conditions (i.e., volume-flux, temperature, density). The initial water vapour content ($m_{H_2O,stack}$ [kg]) in the effluent is determined from annual and/or hourly emission rate inventory data for water vapour. The (known) input stack parameters also include stack-top height $z_{stack}$ in meters [m] agl (above ground level), stack radius $r_{stack}$ [m], stack volume flow rate $\dot{V}_{stack}$ [m³/s], stack/effluent temperature $T_{stack}$ [K], and effluent exit velocity $w_{stack}$ [m/s]. The effluent buoyancy is determined in relation to ambient air information, which can be from sounding data or model generated ambient state variables. The buoyancy-flux immediately above stack-top ($F_0$) is then calculated as the product of effluent buoyant acceleration and the stack volume flow rate ($\dot{V}_{stack}$),

$$F_0 = g \, \frac{\rho_{air} - \rho_{stack}}{\rho_{stack}} \, \dot{V}_{stack} \tag{1}$$

where $g$ [m/s²] is the gravitational acceleration , $\rho_{air}$ [kg/m³] is ambient air density, and $\rho_{stack}$ [kg/m³] is effluent (dry air) density at stack-top.

Briggs (1984) noted that the behaviour of plumes under conditions of low wind speed differed from that in higher wind speeds, and described these two conditions with two different equations, one for "vertical" and the other for "bent-over" plumes. The parcel volume flux as it rises through the plume (which includes the effects of entrainment), $\dot{V}$ [m³/s], is determined based on empirical formulations for buoyant plumes by Briggs (1984),

$$\dot{V}(z) = \begin{cases} 0.791 \, \alpha^{4/3} \, F_0^{1/3} \, z'^{\,5/3} & \text{, vertical.} \\ \pi \, U(z)\beta^2 \, z'^{\,2} & \text{, bent-over.} \end{cases} \tag{2}$$

where $z' = z - z_{stack}$ is the height above stack-top [m], $U$ [m/s] is the horizontal wind speed at $z$ [m], and $\alpha$ and $\beta$ are (dimensionless) empirical coefficients of entrainment. Briggs (1984) formulation made use of the Taylor entrainment hypothesis ("the rate at which ambient air is drawn into the plume is proportional to the velocity shear between the plume and the ambient fluid, and this shear consists mainly of the plume's vertical velocity"). Briggs (1984) recommended (empirical) entrainment coefficients of about $\alpha = 0.8$ and $\beta = 0.6$ for buoyant plumes. The change in effluent plume volume between two adjacent atmospheric heights can be calculated by multiplying the average volume-flux by the transit time between those heights as it rises, $\Delta V(z) = \overline{\dot{V}}(z) \, \Delta t$. The transit time $\Delta t$ [s] can be determined kinematically from parcel vertical velocity and buoyant acceleration at height $z$. Parcel volume $V$ [m³], vertical velocity $w$ [m/s], density $\rho$ [kg/m³], temperature $T$ [K], and buoyant acceleration $a$ [m/s²], are numerically calculated in the algorithm for each consecutive vertical level $z$ (derivations of the formulae presented here are provided in the Supplement section S1). Using these updated parameters, the equivalent vapour pressure of the net amount of water in the parcel is calculated as,

$$e_v(z) = \frac{P_a}{\varepsilon} \, q_v = \frac{P_a}{\varepsilon} \left( \frac{m_{H2O}}{\rho V} \right) \tag{3}$$



where $q_v$ [kg/kg] is vapour mixing ratio, $P_a$ [Pa] is air pressure (equivalent for ambient and parcel air), and $\varepsilon = 0.622$ (Rogers and Yau, 1989). From Iribarne and Godson (1981), the saturation vapour pressure of water [Pa] as a function of temperature of the rising parcel $T$ [K] is given by:

$$e_{sat}(T) = 10^{[-2937.4/T - 4.9283 \, log_{10}T + 25.5471]} \tag{4}$$

In the following, we use simple parcel model parameterizations to estimate the latent heat release/uptake based on the approach described in Rogers and Yau (1989). If the parcel temperature drops below the saturation temperature at a given level, the amount of water mass mixing ratio present in the condensed phase can be derived from the excess vapour pressure above saturation,

$$q_c = max \left[ (e_v - e_{sat}(T)) \frac{\varepsilon}{P_a}, \, 0.0 \right] \tag{5}$$

Note that $q_c$ can be calculated at each model layer by using the total water in the parcel, and that an increase in $q_c$ between two adjacent levels representing the layer mid-points implies that condensation of water mass has occurred between those levels, while a decrease in $q_c$ implies that the evaporation of water mass has occurred between the levels. The corresponding release or uptake of latent heat can be calculated as,

$$\Delta Q_{cond} = L_v \, \Delta (\rho \, V \, q_c) \tag{6}$$

where $L_v$ is the latent heat of condensation. Further, the first law of thermodynamics (at constant pressure $\Delta P = 0$) may be used to determine the change in parcel temperature $\Delta T_{cond}$ resulting from phase change of water (Rogers and Yau, 1989),

$$\Delta T_{cond} = \frac{\Delta Q_{cond}}{M \, C_p} \tag{7}$$

where $C_p = 1004$ J kg$^{-1}$ K$^{-1}$ is specific heat at constant pressure, and $M = \rho \, V$ is the total parcel mass.

As in Briggs (1984), the rate of increase of the volume of the rising air parcel carrying the pollutants is assumed to be solely
due to turbulent mixing between the parcel and the surrounding atmosphere (entrainment), in which case the change of parcel volume with respect to height can be used to estimate the change in mass due to entrainment: $\Delta m_{en}(z) = (\rho_{air}(z)\Delta V(z))$ [kg], where the subscript "air" indicates the ambient, outside-of-plume, conditions at the given height. When the effluent is at a higher temperature than added ambient air mass (i.e., for buoyant plumes $T > T_{air}$), heat is transferred from the effluent to the entrained air,

$$\Delta Q_{en}(z) = - C_p (T - T_{air}) \Delta m_{en}(z) \tag{8}$$

resulting in a corresponding change (decrease) in parcel temperature,

$$\Delta T_{en} = \frac{\Delta Q_{en}}{M \, C_p} \, . \tag{9}$$

Another consideration with regards to entrainment is that the parcel may be rising through air which contains water, in both gaseous ($q_{v,a}$) and liquid ($q_{c,a}$) form, and this water may be entrained during the rise between vertical levels,

$$\Delta m_{\mathrm{H_2O}}(z) = q_* \Delta m_{en}(z) \tag{10}$$



where $q_* = q_{v,a} + q_{c,a}$ is the total water content mixing ratio. The entrained water contributes to the total water within the plume: $m_{H_2O} + \Delta m_{H_2O}$ (with the stack emitted water $m_{H_2O,stack}$ as the initial value). The entrained water can influence parcel condensation or evaporation through adding or removing mass from the condensed phase. If we assume all the water content within the parcel as vapour, the equivalent vapour pressure of the new net amount of water in the parcel can be re-

calculated from Eq. (3). The revised value of $e_v$ can then be used to determine the new value of the condensed phase water within the parcel $q_c$ from Eq. (5). Referring back to Eq. (6), the energy lost or gained due to the entrained water added to the parcel will be *de facto* included in the heat exchange included in the equation.

The **moist** plume rise algorithm is **stratified** in the sense that it performs layered calculations for plume vertical momentum, state variables, and buoyancy. At each height the amount of entrained air and water is determined. Further, the change in

temperature as a result of heat transfer to the entrained air and latent heat release/uptake (due to phase changes of water) is determined. The contributing processes can be summarized as follows,

$$\Delta T(z) = \Delta T_{cond}(z) + \Delta T_{en}(z) = \frac{1}{M\,C_p}\left[\Delta Q_{cond}(z) + \Delta Q_{en}(z)\right] \tag{11}$$

where positive (negative) values of $\Delta T$ indicate increases (decreases) in plume temperature.

The algorithm utilizes an **iterative** solver (Newton-Raphson-Secant method; Oxford, 2014) to calculate parcel temperature,

executing several iterations (up to a user defined maximum iteration number; for our tests 20 to 50 iterations were sufficient) until it converges on a solution for the (equilibrium) parcel temperature at a given layer in the atmosphere. The parcel density is then recalculated from the ideal gas law as a function of the revised parcel temperature and air pressure,

$$\rho(z) = \frac{P_a(z)}{k\,T_v(z)} \tag{12}$$

where water mixing ratios in vapour $q_v(z)$ and condensed $q_c(z)$ phases are accounted for in calculating the updated parcel

density in the virtual temperature term,

$$T_v(z) = T(z)\left[\frac{1 + q_v(z)/\varepsilon}{1 + q_v(z)} - q_c(z)\right] \approx T(z)\left(1 + 0.61 q_v(z) - q_c(z)\right) \tag{13}$$

Note that the addition of condensed water further modifies parcel buoyancy (see Chapter 3, Stull, 2017). The updated parcel density is then compared to ambient air density. If solution results in positive buoyancy (that is, the parcel density is still below that of the ambient air), plume continues to rise to the next vertical level up. These layered calculations are repeated up to

the vertical level at which the plume buoyancy is either zero or negative ($\rho(z) \geq \rho_{air}(z)$). The height of this vertical level is then taken as the final plume height. Finally, the plume vertical spread is determined from the plume rise above the stack height $\Delta h$ and the emitted mass is uniformly distributed in the vertical between plume bottom and top determined following the commonly used method from Briggs (1975),

$$h_t = h_s + 1.5\,\Delta h,$$
$$h_b = h_s + 0.5\,\Delta h, \tag{14}$$

where $h_s$, $h_t$ and $h_b$ are stack top, plume top and plume bottom heights, respectively.



Our new plume rise algorithm **PRISM** (Plume-Rise-Iterative-Stratified-Moist) is essentially a 1D model (with user defined resolutions and parameters) that can be run as standalone or embedded within a host 3D model (in this case GEM-MACH) as a sub-grid parameterization scheme. It takes stack parameters (e.g., volume flow rate, temperature, water content) and ambient air state variables as input information and performs high (vertical) resolution layered calculations of parcel buoyancy driven

rise. At each height, the algorithm calculates the change in parcel temperature (and corresponding change in density) as it rises, expands, and mixes with the ambient air, while taking into account the effects of latent heat uptake/release due to phase changes of within parcel water content. Note that the release or absorption of latent heat due to condensation or evaporation of water in the parcel may serve to decrease or increase parcel buoyancy, depending on ambient conditions such as the temperature profile and ambient water content. See Supplement Section S1 for algorithm details and the corresponding discrete numerical

formulations.

## 2.2 Model Description and Setup

The Global Environmental Multiscale – Modelling Air-quality and Chemistry (GEM-MACH) is Environment and Climate Change Canada's (ECCC) air quality prediction model (Moran et al., 2010). GEM-MACH is an online air-quality and chemical transport model, which resides within the Global Environmental Multiscale (GEM) numerical weather prediction model (Côté

et al., 1998b,a; Girard et al., 2014). The GEM meteorological model and its components have been extensively evaluated elsewhere in the literature (e.g. Côté et al., 1998b; Bélair et al., 2003b,a; Li and Barker, 2005; Milbrandt and Yau, 2005a,b; Fillion et al., 2010; Girard et al., 2014; Milbrandt and Morrison, 2016). In addition to the GEM weather prediction model, GEM-MACH includes an atmospheric chemistry module (Moran et al., 2010) with gas and particle process representation. GEM-MACH is used here in its fully coupled configuration – i.e., the model's particulate matter is allowed to modify the

meteorological predictions through the aerosol direct and indirect effects (Makar et al., 2015a,b; Gong et al., 2015). For a recent evaluation of GEM-MACH's performance see Makar et al. (2021), and also see Fathi et al. (2021) for a comprehensive discussion on tracer mass-budget and transport in GEM-MACH. For this work a nested configuration for GEM-MACH was used with a parent domain covering North America at 10 km resolution and a nested high-resolution domain with 2.5 km grid spacing over the Canadian provinces of Alberta and Saskatchewan, including the Athabasca Oil Sands regions (see Figure 1a).

This region has been characterized by an extensive effort to improve emissions inventory inputs for regional model simulations (Zhang et al., 2018) and hence is ideal for tests of plume rise algorithms, the results we show are generic, and applicable to all other cases of plume rise driven by combustion sources of heat. The details of the GEM-MACH model configuration used in this work appear in Appendix Table A1.

Note that the initial implementation of the plume rise in GEM-MACH utilized Briggs (1984) empirical formulation based on

source parameters and estimates of atmospheric stability at stack top (Moran et al., 2010). Later, plume rise in GEM-MACH based on Briggs (1984) was further refined to include layered calculation of plume residual buoyancy above the inversion height, as described in Akingunola et al. (2018). For this work we configured the GEM-MACH model at high-resolution (2.5 km grid spacing) to perform two sets of retrospective air-quality model simulations with different plume rise options:





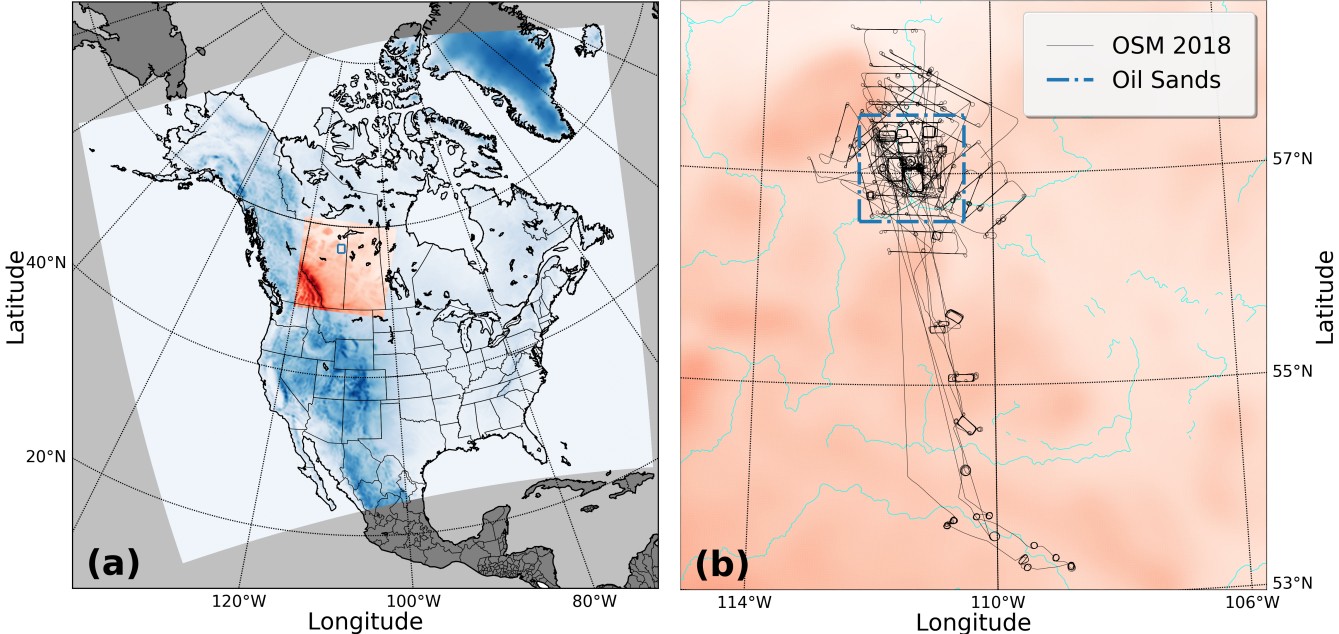

**Figure 1.** (a) GEM-MACH model nesting configuration with a parent domain at 10 km resolution over North America (blue shaded area), and nested domain at 2.5 km resolution (red shaded area) over Alberta and Saskatchewan provinces. The approximate perimeter of Athabasca Oil Sands is shown with a blue rectangle. (b) Oil sands region within the 2.5 km domain is depicted with flight tracks (dark) from the OSM 2018 aircraft campaign overlaid on the map. The region encompassing the surface mining facilities of the Athabasca Oil Sands is shown with a blue dashed line. Most of the region's $SO_2$ emissions occur from large stacks associated with upgrading of bitumen at surface mining facilities in within the dashed line.

(a) original GEM-MACH plume rise based on Akingunola et al. (2018) hereafter referred to as GM-orig, and (b) PRISM as
described in this work (Section 2.1) hereafter referred to as GM-PRISM.

## 2.3 Aircraft Campaign

During the OSM 2018 campaign (April to July), aircraft-based measurements of environmental variables (meteorology, pollutant concentrations) were conducted over Canadian Oil Sands (OS) (ECCC, 2018). Figure 1 (b) shows flight tracks taken by the aircraft during the OSM 2018 campaign over the OS region. The aircraft conducted several flights during different
days and times, through April to July 2018, including single screen flights tens of kilometres downwind of OS facilities and box flights around the facilities at near range. The designation "box flight" refers to a flight pattern during which the aircraft would fly along closed loops around a specific emitting facility at several consecutive altitudes while making measurements of



environmental variables, and were specifically designed to capture emissions from individual facilities. Aircraft measured data during box flights were converted into source emission rates, through flux estimations and mass-balance calculations utilizing the TERRA algorithm described in Gordon et al. (2015). This was done for several emitted species such as $SO_2$, $NO_x$ and $CO_2$. As discussed in the next section, aircraft-based estimates, emission inventory data, and Continuous Emissions Monitoring System (CEMS) data for $NO_x$ were used to derive $NO_x$ to $CO_2$ emission rate ratio, in turn used to estimate the water emissions rate.

Here, we also used aircraft measurements of $SO_2$ concentrations downwind of several oil sands facilities (CNRL, Syncrude and Suncor) to determine observed plume heights and evaluate our model predicted plume rise against these observations. For our analysis, we considered aircraft data from box flights where measurements were made just a few kilometres downwind or upwind of emission sources. This was done to avoid flights which included a large long range transport path/time of emitted pollutant to the point of measurement, so that the observed plumes would be a better representation of emission and plume rise condition at the stack locations. We focused on $SO_2$ as the emitted pollutant, since it is a primary emitted pollutant (i.e., not generally generated due to photo-chemical reactions in the atmosphere), and due to the availability of Continuous Emissions Monitoring (CEMS) based direct observations of $SO_2$ within emitting stacks. $SO_2$ in Oil Sands (OS) is mainly emitted from large high-temperature stack sources (over 90%, unlike $NO_2$ which is only about 40% emitted from large stacks, Zhang et al., 2018) with low background levels from other sources, making $SO_2$ a good indicator for buoyant plumes and suitable for our study of plume rise parameterization.

## 2.4 Input Emission rates and Source Parameters

Water vapour ($H_2O$) and Carbon dioxide ($CO_2$) emission rates from sources within the OS facilities are neither reported in emission inventories such as National Pollutant Release Inventory (NPRI: ECCC, 2023) nor as part of Continuous Emission Monitoring System (CEMS). However, their emissions are correlated with fuel combustion as part of OS productions/activities; $CO_2$ and $NO_x$ emissions are related to synthetic crude oil production at OS (Liggio et al., 2019). For this work $NO_x$ emission rates, which are reported in NPRI and CEMS datasets, are used as a proxy for estimating $CO_2$ emission rates, and the corresponding water emission rates determined from combustion reaction stoichiometry. The stoichiometry of the relative amounts of water to $CO_2$ emitted for the given fuel thus provide an estimate of the water emitted due to combustion. Wren et al. (2023) calculated the average ratios of $CO_2$ to $NO_x$ emission rates from OSM 2018 aircraft campaign data for individual OS facilities and source types (e.g., stack, area). For this work, $CO_2$:$NO_x$ ratios estimated by Wren et al. (2023) for the stack sources were used in turn to estimate $CO_2$ emission rates from $NO_x$ reported in NPRI and CEMS. $CO_2$ and $H_2O$ are primarily generated from combustion of natural gas, with Methane ($CH_4$) as its main component, in OS production operations,

$$CH_4 + 2\,O_2 \longrightarrow CO_2 + 2\,H_2O \tag{15}$$

Therefore for every mole $CO_2$, 2 moles of $H_2O$ are emitted due to combustion. Accordingly, a stoichiometric ratio of 1:2 of $CO_2$ to $H_2O$ can be used to estimate $H_2O$ emissions levels, as was done for this work. $H_2O$ emissions were then calculated from NPRI and/or CEMS reported $NO_x$ emission rates, based on source specific $CO_2$ to $NO_x$ ratios. The Continuous Emissions



Monitoring (CEMS) hourly data was available for $SO_2$ and $NO_x$ for only two of the OS Suncor stack sources, and for $SO_2$ for other facilities/stacks, for the period corresponding to the aircraft study. Canadian emissions reporting requirements for NPRI reporting for large stacks are for annual totals. Therefore, the hourly $NO_x$, and consequently hourly $H_2O$, for the rest of the facilities were estimated from NPRI annual emissions data. CEMS hourly data for stack parameters (e.g., exit temperature,

flow rate) and $SO_2$ emission rates were available for April to July 2018, partially overlapping the period of our 6-month run simulations from February to July 2018, and were used in the simulations for the same period. We note that the estimation of stack water emissions is a required input for our algorithm - the methodology demonstrated here is easily expandable to other combustion stack sources. Knowledge of the fuel type is required, with different fuels having different amounts of water produced per carbon atom combusted — i.e. Eq. (15) depends on the fuel used for generating heat for stack emissions. As we

will discuss below, the accuracy of the stack emissions and the consequent estimates of water emissions, have a key impact on the accuracy of our plume rise algorithm. Note that we used the estimates of combustion-generated water, as described above, in our standalone offline simulations with PRISM for the specific stack sources for which the following information were available: (a) reported $NO_x$ and $SO_2$ emission rates, (b) facility-specific estimates (aircraft-based) of $CO_2$ to $NO_x$ emission ratios. Such source-emission information were not available for the majority of the stack sources within our large-

scale GEM-MACH modelling domain (10km resolution domain over North America, 2.5km resolution domain over Alberta and Saskatchewan). Therefore, in our GEM-MACH simulations with PRISM (GM-PRISM) we solely accounted for the impact of entrained water from ambient air on plume buoyancy and final rise.

## 2.5  WBEA surface monitoring network

We evaluated model performance in terms of surface concentrations of $SO_2$ against air quality observations from 21 WBEA

(Wood Buffalo Environmental Association) continuous surface motoring stations in Alberta. Here, we focus on $SO_2$ as a primary emitted pollutant. Given that $SO_2$ is mainly emitted from large smokestacks at OS region (over 90%, Zhang et al., 2018), this makes it more relevant for our purposes: evaluating the plume rise parameterization for buoyant sources. We analyzed the hourly WBEA data from February to July 2018, against model generated fields for the same period.

## 2.6  Case Studies

We considered a simulation period for 2018 over the Canadian Oil Sands (OS). This period overlaps with the OSM 2018 aircraft campaign over the oil sands region between April to July of 2018. The aircraft campaign is discussed in Section 2.3. We performed high-resolution (2.5km grid spacing) air quality simulations with the GEM-MACH model, focusing on the Athabasca oil sands region. Our new plume rise algorithm PRISM was implemented with the high-resolution GEM-MACH simulations (GM-PRISM) for a 6-month model run (February to July 2018 inclusive), and was compared to simulations carried

out with the previous scheme (GM-orig) (the latter lacking full stratified calculations of plume buoyancy and water latent heat release/uptake, Akingunola et al., 2018). Model output data from the simulation period were compared to data from the WBEA surface monitoring network for the region, and to aircraft observations from the OSM 2018 campaign. In our analysis, we focused on plumes emitted from the three main (largest) $SO_2$-emitting facilities: Syncrude, Suncor, and CNRL. We compared





model-generated $SO_2$ fields to aircraft $SO_2$ measurements from 11 box flights around the three facilities of interest. The aircraft

data allow us to directly compare model and observed $SO_2$ plume heights and thus provide a direct estimate of plume rise accuracy (the surface monitoring network data, the analysis of which follows the plume height evaluation, allow us to estimate the effect of the changes on surface $SO_2$ concentration predictions). Four of these flights were conducted in April and May of 2018 (2 flights each months) and the rest (7 flights) were conducted in June of 2018. Hence, while April in this region is snow-covered and represents emissions under winter conditions, the majority of available aircraft data were for the summertime.

Aircraft measured/interpolated wind and $SO_2$ data were used to determine plume origins (emission sources). We note that the box flights were designed with the intent of sampling plumes from specific facilities; combined with the aircraft wind speed and direction data, the emissions associated with the source within an enclosing box flight can be distinguished from other sources in the region. Flight planning included wind forecasts that allowed box flights to avoid conditions under which a plume from one facility impacted the air above another facility. $SO_2$ data recorded during the segments of the flights corresponding

to model output data were analyzed to determine plume centre heights (height of the maximum observed concentrations). The observed heights were compared to model-predicted plume heights using the two plume rise algorithms GM-orig and GM-PRISM. The results of these evaluations and comparisons follow.

## 3    Results and Discussion

### 3.1    Model Sensitivity to Plume Rise Parameterization

We investigated the impact of within-plume combustion generated water on the neutral buoyancy height of the effluents from high temperature stacks using PRISM. Figure 2 shows the dependence of plume final height on stack temperature and the amount of water released within the plume parcel, for an idealized case with dry adiabatic lapse rate. The range of stack temperatures and water emissions are taken from the corresponding reported parameters for the stacks of interest for three Oil Sands (OS) facilities: CNRL, Suncor, and Syncrude. Note that initial in-plume water vapour was limited to values less

than or equal to the saturation level dictated by the saturation vapour pressure for each given stack exit gas temperature (note the cutaway in the surface plot in Fig. 2, and that the high temperatures allow for much higher water content than might be found at ambient temperature conditions). The dependence on stack exit temperature is evident from the results shown in Fig. 2 – i.e., higher stack temperature corresponds to higher plume parcel (initial) buoyancy and the resulting increase in the final height reached by the plume parcel (neutral buoyancy height). The other interesting observation is the stronger dependence

on the amount of emitted water vapour. Our results show the significant impact of latent heat exchange due to phase changes of within-plume water on plume rise (Fig. 2). The net release of latent heat as the water vapour condenses within the rising plume modulates plume parcel buoyancy significantly, resulting in up to 500 m additional rise for the case shown in Fig. 2; compare the plume height values (vertical axis, Fig. 2a) for zero water emissions to those at the maximum water emissions. The dependence trends (the cross-sectional trends in Fig. 2) reveal that plume neutral buoyancy height is impacted by moist

thermodynamics more significantly than by parcel initial temperature.





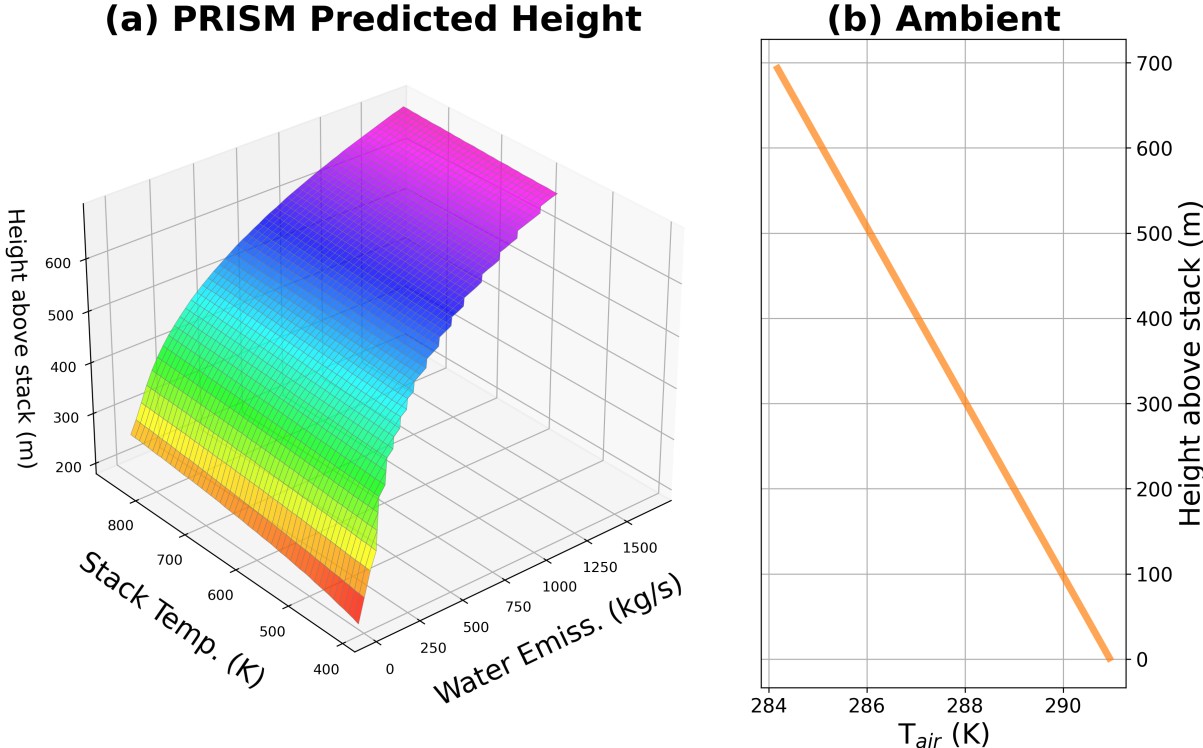

**Figure 2.** (a) PRISM predicted final plume rise for an idealized case as a function of stack temperature and emitted water. (b) the idealized ambient profile for air temperature (with dry adiabatic lapse rate) is shown as a function of height. Plume neutral buoyancy height shows stronger dependence on initial in-plume water vapour than stack temperature, resulting in up to 500 m additional rise for the range shown.

We also investigate the impact of moist thermodynamics on plume rise for realistic cases with more complex atmospheric vertical structures. Using the GEM numerical weather model (see Table A1) at high-resolution (2.5km grid spacing), we generated meteorological fields (wind, ambient air density, temperature, vapour and liquid water mixing ratios) for the 2018 aircraft campaign over the Oil Sands region. We used the model generated meteorological fields (vertical profiles) corresponding to

the period of 11 box flights around three OS facilities (Suncor, CNRL, Syncrude) as input in PRISM. Further, we used stack parameters (temperature, volume flow rate, water emission rate) for high temperature stacks within these three facilities to model plume rise using PRISM (off-line – i.e., not embedded within the GEM-MACH 3D model). Figure 3 shows the results for a case on 6 June 2018 for the Syncrude main stack. Plume parcel temperature (T), density ($\rho$), water vapour mixing ratio ($q_v$), condensed water mixing ratio ($q_c$), and parcel rise speed are compared to environmental parameters as a function of

height in Fig. 3. The PRISM predicted parcel state variables are shown for four different rise cases: vertical and bent-over rise, with and without in-plume water. The without in-plume water (dry) rise cases, illustrated with dashed curves, show how parcel temperature drops (and density increases) as the rising parcel mixes with the ambient air (through entrainment), until it reaches the neutral buoyancy height (height at which parcel density approach ambient air density). Note that for the bent-over





plume rise, the parcel volume flux is a function of the horizontal wind speed $U$ (cross-wind shown with a blue curve in Fig.
3e) according to Eq. (2). Consequently, even in the presence of mild cross-winds (2 m/s to 5 m/s), expansion (due to entrainment) and buoyancy reduction rate is higher for the bent-over rise than the vertical rise, and therefor the parcel reaches neutral buoyancy at lower altitudes (Fig. 3). PRISM performs both (vertical and bent-over) calculations for each plume rise case, and following Briggs (1984) chooses the final rise calculated by the one resulting in higher buoyancy reduction as a function of height. The impact of latent heat exchange can be seen for the moist plume rise cases, shown with solid curves in Fig. 3. The
condensation of in-plume water (and the resulting latent heat release) prolongs parcel buoyancy for both rise types (vertical and bent-over), resulting in higher final rise compared to dry cases. Note the difference in condensed water ($q_c$) vertical profiles for the bent-over (green) and vertical (orange) rise types in Fig. 3d. Water condenses faster (and at lower altitudes) for the bent-over rise, but it is short-lived compared to the vertical rise. The corresponding impact on parcel temperature (T) and density ($\rho$) can also be seen in Fig. 3a,b: parcel temperature drops (and parcel density increases) with height at a lower rate for the period of
latent heat release (compared to rise with no latent heat exchange), and as a result the parcel state variables approach ambient values at much higher altitudes. Note that the height at which parcel plume density approaches ambient air density, within an acceptable level of accuracy (defined as a convergence criteria of the difference between parcel and ambient air density relative to the ambient air density being below a threshold), is taken as plume neutral buoyancy height. Under most convective conditions (and in the absence of strong inversions), parcel density tends to approach ambient air density asymptotically (see Fig.
3b). PRISM can be configured with different density convergence criteria ($\rho_{conv}$) in terms of percentage difference between parcel and ambient air density: $\rho_{conv} = |(\rho - \rho_{air})|/\rho_{air} \times 100$. The height at which the difference between parcel density and ambient air density falls within $\rho_{conv}$, parcel is assumed neutrally buoyant and the rise terminates. We performed tests with $\rho_{conv}$ ranging between 0.1% to 0.5%, and found $\rho_{conv} = 0.3\%$ to be the optimal convergence criteria for the majority of the cases we considered, by comparing plume rise estimates to aircraft-observed plume heights. We note that the choice of
$\rho_{conv}$ depends on the numerical accuracy of the calculations and the vertical resolution at which plume buoyancy reduction is calculated. The results shown in Fig. (3) are from calculations at 1m resolution. Our tests with different resolutions up to 10m resolution have shown optimal performance with $\rho_{conv}$ between 0.3% and 0.5%. We also note that the plume rise algorithm is sensitive to input information such as stack exit temperature, and depending on the confidence level of input parameters, the convergence criteria can be either strict or relaxed.

Finally, we performed two sets of retroactive simulations with the GEM-MACH model, with the original plume rise algorithm (GM-orig) and with PRISM embedded within GEM-MACH (GM-PRISM). Model output from the two sets of simulations were compared for a 6 month period between February to July 2018. Output data were divided in two groups, the wintertime (including months of February, March, and April) and summertime (including months of May, June, and July). This was done in order to investigate model sensitivity to the two different plume rise parameterizations for two general sets
of conditions, the cold and more stable atmosphere during the wintertime and the warmer and less stable atmosphere during the summertime. The separation of the simulations into the two seasons also allows us to examine the effect of emissions data accuracy on plume rise calculations: we note that CEMS source parameter and emissions data were available only for April to July 2018 (excluding the months of February and March in the wintertime). The average $SO_2$ surface concentrations for





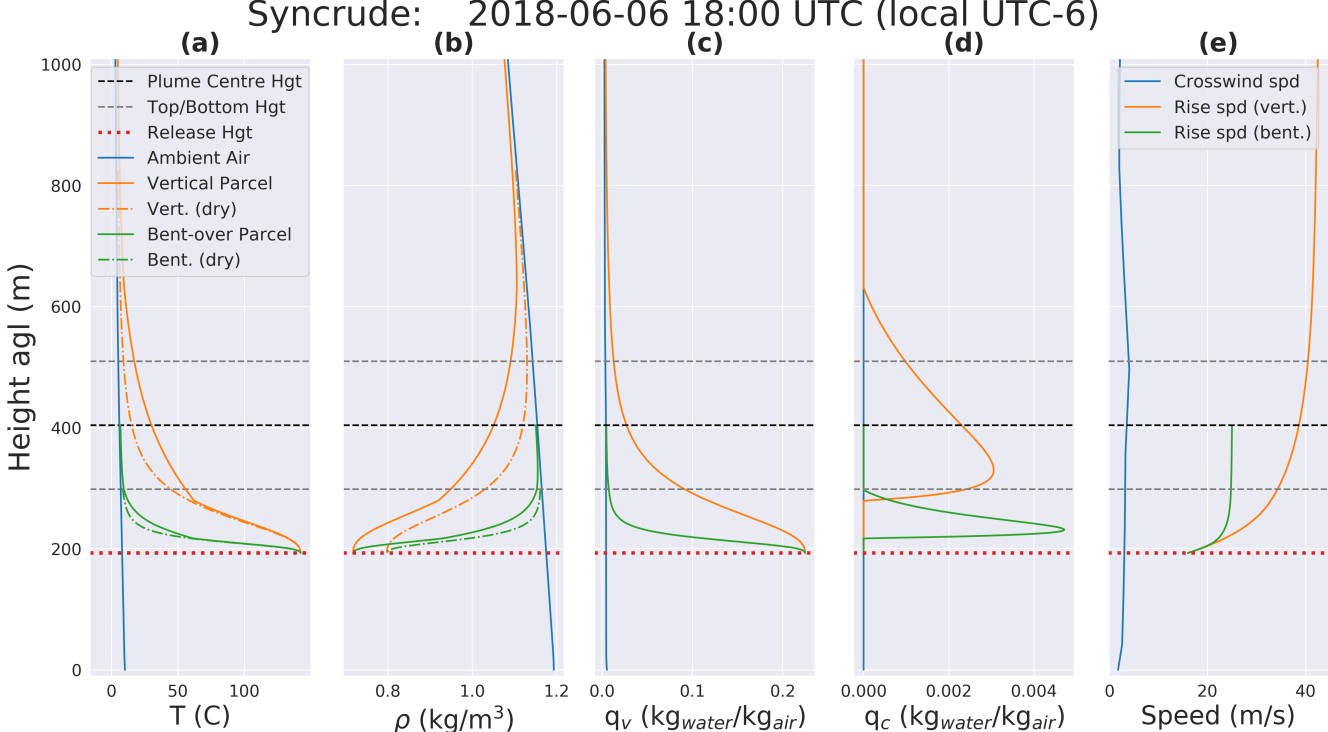

**Figure 3.** PRISM predicted parameters for a rising plume parcel are compared to ambient conditions for four different cases, vertical and bent-over rise (moist and dry), for the main stack at OS Syncrude facility on 6 June 2018 at 18 UTC. Parcel (a) temperature, (b) density, (c) water vapour mixing ratio, (d) condensed water mixing ratio , (e) parcel rise speed and horizontal wind speed $U$ (crosswind) are shown.

GM-PRISM summertime simulations, with $\rho_{conv} = 0.3\%$, are shown in Figure 4 (a) and (b) for the Oil Sands Region sub-

domain and the entire high-resolution domain, respectively. Figure 4 (c,d) show GM-PRISM normalized mean bias (NMB) in % relative to GM-orig simulations for surface $SO_2$. The confidence ratios at 90% confidence level (CR90, see Makar et al., 2021) were also calculated between surface concentrations generated by the two simulations and, are depicted in Fig. 4 (e,f). The confidence ratio values $\geq 1$ are indicative of a statistically significant difference between the GM-PRISM and GM-orig simulations at the specified confidence level (here 90%). The highest values of CR (e.g. 2 and above) are located close to

sources of $SO_2$, such as the Oil Sands sources, as well as other sources located to the south and west of the Oil Sands Region (Fig. 4 e, f). That is, the impact of the revised parameterization is the strongest close to the sources. We note that due to the lack of sufficient information (e.g., source specific $CO_2$ to NOx emission rate ratios) for reliably estimating the amount of combustion generated water mass for the hundreds of emission sources within the large-scale modelling domain, the emissions of combustion water were unavailable for GEM-MACH simulations, and the differences between the algorithms are due to

the entrainment of ambient water into "dry" combustion plumes and the stratified calculation of plume buoyancy in PRISM. Nevertheless, differences can be seen for all large stack sources of $SO_2$ within the domain, showing the impact of the revised





**Figure 4.** Average surface $SO_2$ concentrations for the summertime period (May, June, and July of 2018) generated by GM-PRISM simulations with $\rho_{conv} = 0.3\%$ shown for (a) oil sands region and (b) the entire domain. (c,d) normalized mean bias (NMB) in % relative to GM-orig simulations for the same period. (e,f) confidence ratio at 90% confidence level (CR90).





algorithm on $SO_2$ even in the event that water emissions are not available; entrained water interacts with emitted parcels and may have a significant impact on plume rise and $SO_2$ dispersion. CR90$\geq$ 1 values near large stack sources clearly demonstrate that the plume rise algorithms predicted different plume heights at source locations, resulting in different vertical distribution

of the $SO_2$ plumes and significant differences at the surface. These differences become less pronounced farther away from the emission sources, though some regions of significant differences (also significant at lower confidence levels – e.g., CR80$\geq$ 1, CR85$\geq$ 1) can occur far downwind of the sources (e.g. northern Saskatchewan; pink region in middle-right of Fig. 4 f). The downwind differences demonstrate the change in the direction and the range of the transport of the emitted $SO_2$ mass. This is a direct result of the difference in rise parameterization, due to the plumes rising to different altitude levels with dissimilar

flow regimes (e.g., wind speed and direction, strength of turbulence). Similarly for the wintertime, the difference between GM-PRISM and GM-orig simulated surface $SO_2$ were pronounced near the OS sources (see Fig. S1 for wintertime comparisons, which were similar to those displayed here for summer conditions).

The GM-orig algorithm parameterizes the plume rise based on flux reduction calculations as a function of atmospheric stability (Akingunola et al., 2018). Whereas, the GM-PRISM algorithm performs direct flux reduction calculations at each

vertical level while accounting for heating/cooling due to phase changes of water. Consequently, the GM-PRISM algorithm is highly sensitive to input stack parameters and in-plume water mass data. We note that hourly CEMS data (direct measurements) of source parameters (eg., effluent exit temperature and volume flow rate) were only available for the period between April to July 2018 (summertime) as input into model simulations. The input stack parameters for the months of February and March (2/3 of wintertime) were based on annual total reporting to the Canadian National Pollutant Release Inventory (NPRI). The

reported stack parameters are "optimal" values for a given stack, but may not correspond to hour - to - hour variations. The winter stack parameter estimates are largely indirect (based on other factors such as design parameters of the stack) at low temporal resolutions (i.e. based on annual total emissions data; AER, 2022). This adds further uncertainty for wintertime evaluations of GM-PRISM simulations.

## 3.2  Plume rise evaluation against aircraft observed $SO_2$ plumes

Model plume height predictions by GM-orig and GM-PRISM corresponding to 11 box flights during the OSM 2018 campaign were evaluated against aircraft observations for $SO_2$ plumes. Aircraft measurements of wind and concentration fields at several altitude levels around the major $SO_2$ emitting OS facilities CNRL, Syncrude, and Suncor were analyzed to determine the source stack of each observed plume. Note that ambient atmospheric meteorological variables were extracted from the GEM-MACH simulations and used for meteorological inputs for the algorithm. Plume centres for each flight case were identified

and their altitudes estimated from the interpolated concentration data (see Figure S2). These observed plume heights were then compared to plume height predictions by GM-orig and GM-PRISM ($\rho_{conv} = 0.3\%$) simulations for the corresponding times and locations. Figure 5 shows the comparisons between hourly model predicted plume heights at the stack location and aircraft measured vertical profiles of $SO_2$ concentrations corresponding the same model hour. The flight strategy for these box flights was to encircle the facility, starting the aircraft flights around the facility close to the surface, and increasing in altitude

as the aircraft flew around the facility: a box-shaped "spiral" flight pattern, gradually increasing in height (see Fig S2). High





**Figure 5.** Predictions of plume height in GM-orig and GM-PRISM simulations compared to OSM 2018 aircraft observations for the eleven case studies. For each hour of the flight, aircraft observed vertical profiles of concentration are shown as density maps (white to blue) up to the height visited by the aircraft by that hour. Concentrations are shown as shaded blue regions, which have been normalized to the maximum concentration encountered during the flight.

concentration $SO_2$ on a given pass around the facility was taken as a tentative plume height on each pass as the aircraft rose in altitude. However, the highest concentration encountered during the entire set of passes was used to represent the plume height, with lower concentrations encountered during the course of the flight representing either the edges of a rising plume or smaller concentration plumes due to other sources within the facility and region (Fig. S2). In some cases, during the course

of a flight, the apparent equilibrium plume height (determined from the highest concentration encountered during a given pass around the facility) changed, possibly reflecting an ongoing change in plume height due to changing atmospheric conditions (Fathi et al., 2021). That is, the top of the plume could be distinguished close to the surface, and then again at a higher level



on a subsequent higher altitude pass of the aircraft, suggesting either a rising plume during the course of the study, or multiple layers of $SO_2$ within the box domain. The final estimation of the plume height in these cases was the location of the highest

concentration encountered during the course of the flight. In Figure 5, we show the normalized concentration of $SO_2$ measured at each hour by the aircraft, indicating the height of the observed plume using the maximum concentration at each time. For flights 4, 7, 9, 10, 17, and 21, the plume height increased during the course of the flight. In flights 6, 8, 11, 19 and 20, the plume height remained stable. In some of the cases where the plume height increased, the estimate of the observed height at the first hour (lowest elevation passes around the facility) is highly uncertain, since the flight had yet to reach a height in

which the entire vertical extent of the plume was sampled. Flights 7, 9, 10, 17, 21 are examples where the aircraft sampling during the initial hour may have not yet reached sufficient heights to sample the entire plume. The maximum concentration recorded by the aircraft during each hour was then compared against hour by hour model-predicted plume heights. Model values for the plume height at each hour are shown in symbols in Figure 5 (grey lines and circles – GM-orig, orange lines and squares – GM-PRISM), and the upper and lower extent of the simulated plume via Eq. (14) is shown as a grey (GM-orig) or

orange (GM-PRISM) shaded regions. Note that most of these flights were conducted during local noon and afternoon hours under convective conditions (see Figure S3 for model vs. aircraft observed temperature profiles). Therefore, it is reasonable to assume a temporal variation in the vertical mixing of the observed plumes. Such temporal trends were captured by both GM-orig and GM-PRISM simulations as can be seen in Figure 5.

The GM-PRISM showed a significant improvement relative to the GM-orig in eight of the eleven flights (Flights 7, 8, 9, 11,

17, 19, 20, 21). For these cases, the GM-orig was shown to overestimate the plume height by up to a kilometer (e.g., Flight 8), while the distance between measured and modelled plume heights is greatly reduced with GM-PRISM simulations. For two flights (Flights 6 and 10), the two algorithms produced similar plume heights, and for one flight (Flight 4), both approaches resulted in a considerable overestimate of plume height (possibly due to a positive bias in model temperatures, discussed later). Figure 6 compares GM-orig and GM-PRISM simulated plume maximum concentration heights (GM-orig: grey line,

GM-PRISM: orange line) to the median of max. conc. heights observed during flight/sampling time (blue solid line). The tendency of GM-orig to overestimate plume height can clearly be seen, as can the general overall improvement in plume height with GM-PRISM. The summary values for normalized mean bias (NMB) and normalized root mean square error (NRMSE) in the plume heights are shown in Figure 6; the use of GM-PRISM has substantially reduced the magnitudes of both error metrics, with the NMB decreasing from 60% to 10%, and the NRMSE being halved. The new parameterization thus provides

a clear improvement in plume height estimate compared to the previous algorithm, indicating that the stratified calculation of plume buoyancy and latent heat exchange associated with in-plume water has a significant impact on plume rise. The impact of moist thermodynamics alone can be seen by comparing to PRISM estimates with no in-plume water (dry), shown with dashed orange line in Fig. 6. Dry PRISM underestimated plume neutral buoyancy by 30% in terms of NMB against the observed heights. Which is corrected by 20% when in-plume water thermodynamics are accounted for (Fig. 6).

We note that GM-PRISM over-predictions for Flight 4 (wintertime) are partially due to a positive bias of a few degrees Celsius in model temperatures relative to aircraft measurements (see Figure S3). When this temperature bias is corrected for, GM-PRISM plume height predictions are further improved with NMB=0.0 and NRMSE=0.15. This demonstrates the





**Figure 6.** Aircraft observed SO$_2$ plume heights shown with blue solid line and diamonds. Plume height predictions by GM-orig (grey circles) and GM-PRISM (orange squares) are compared with aircraft observed heights (flight median). Results are shown as normalized mean bias (nmb) and normalized root mean squared error (nrmse). Dashed line show PRISM predictions with plume water content set to zero (dry).

sensitivity of the new parameterization (GM-PRISM) to input ambient temperature profiles. Over/under-predictions, similar to the case of Flight 4, can potentially be related to model temperature biases, though insufficiently precise stack parameter data may also play a role, as discussed above. By using aircraft observed temperature (vertical) profiles as input into the PRISM, we were able to confirm this effect for Flight 4 (reduction in error parameters). We note that, for the current work we had wintertime aircraft data from only two flights (4 and 6), while a larger observational data set is needed for a more comprehensive investigation of such effects. Note that ambient air data required as input for the PRISM algorithm include horizontal wind speed, air density, air pressure, and water content (vapour, liquid, ice) mixing ratio in addition to temperature profiles. For the Flight 4 example only temperature profiles were replaced with aircraft-observed temperatures and the rest of the ambient air input data were from the GEM-MACH model output.



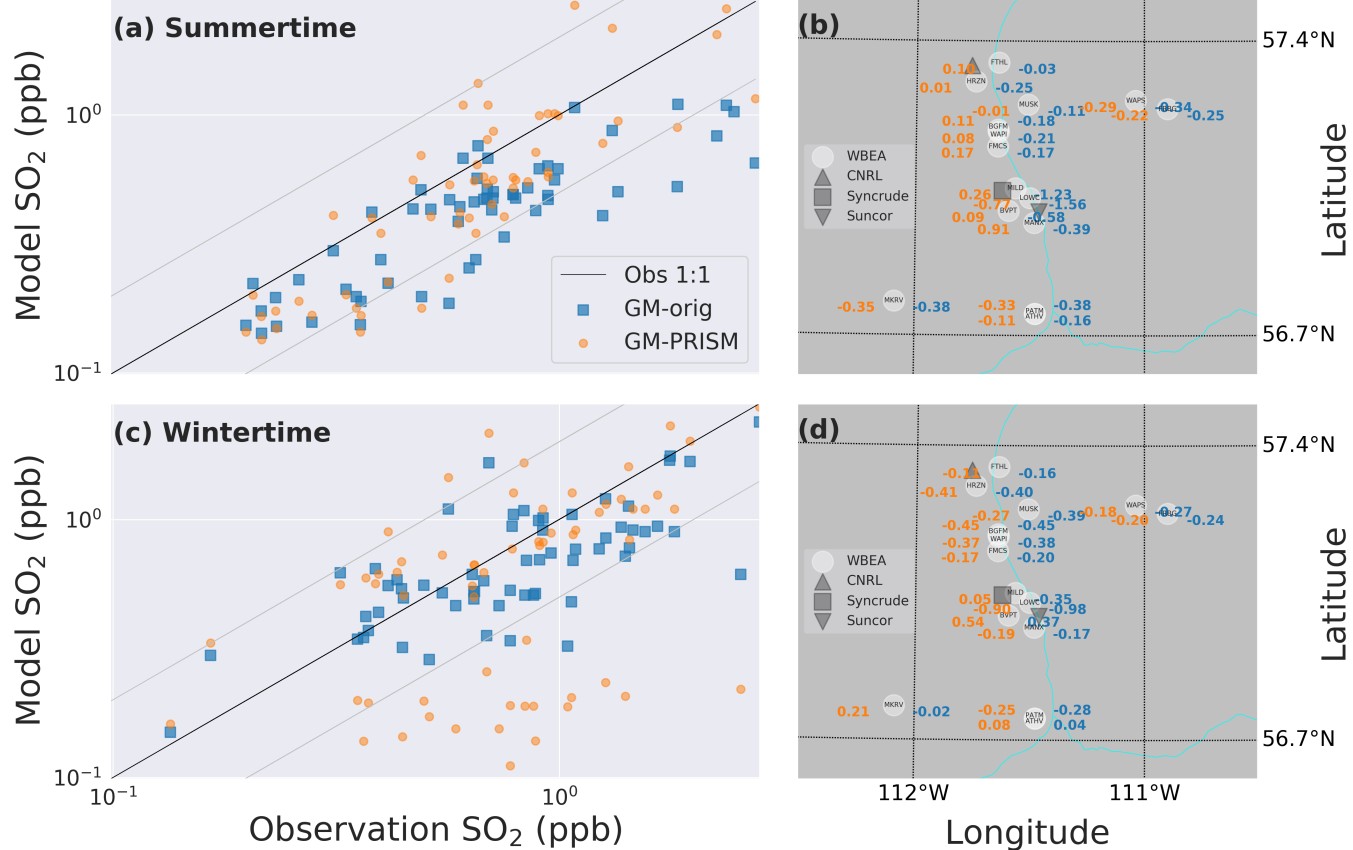

**Figure 7.** Evaluations against WBEA monthly average surface $SO_2$ observations. Comparisons for (a, b) summertime, and (c, d) wintertime are shown. (b, d) show model mean bias in ppb (GM-orig:blue, GM-PRISM:orange) against observations at WBEA stations on the map of OS region for summer and winter, respectively. Also shown in (b) and (d) are the locations for WBEA continuous monitoring stations (white circles) and OS facilities Syncrude (square), Suncor (downward triangle) and CNRL (upward triangle). WBEA station IDs are noted on corresponding white circles.

### 3.3 The impact of moist plume rise on regional surface $SO_2$ concentrations

Evaluations against the WBEA continuous monitoring network confirms the results against aircraft observed $SO_2$ plumes, and shows the substantial impact of moist plume rise on downwind $SO_2$ concentrations, with GM-PRISM improving prediction
of surface $SO_2$ concentration relative to GM-orig predictions for the study period. Figure 7 shows the evaluation of monthly average surface $SO_2$ produced by the model when making use of the two plume rise calculations versus observations at WBEA continuous monitoring stations in the oil sands region. Comparisons are shown for the summertime (May, June, July; when CEMS data were available) in Fig. 7 (a), (b) and the wintertime (February, March, April; when CEMS data were mostly unavailable) in Fig. 7 (c), (d). Fig. 7 (b) and (d) show $SO_2$ mean biases by GM-orig and GM-PRISM (with $\rho_{conv} = 0.3\%$) at the




**Table 1.** Statistical comparison of average monthly $SO_2$ surface concentrations against WBEA continuous monitoring data with GM-orig and GM-PRISM (with $\rho_{conv} = 0.3\%$) simulations for the period from February to July 2018. **R** is the correlation coefficient, **FAC2** is the fraction of predictions within a factor of 2 of observations, **NMB** is the normalized mean bias, and **RMSE** is the root mean squared error.

| Statistics | Summertime | | Wintertime | | Full 6-month | |
|---|---|---|---|---|---|---|
| | GM-orig | GM-PRISM | GM-orig | GM-PRISM | GM-orig | GM-PRISM |
| R | 0.78 | 0.80 | 0.70 | 0.57 | 0.70 | 0.69 |
| FAC2 | 0.73 | 0.84 | 0.90 | 0.63 | 0.82 | 0.74 |
| NMB | -0.42 | -0.06 | -0.19 | -0.16 | -0.30 | -0.11 |
| RMSE | 0.52 | 0.45 | 0.43 | 0.57 | 0.48 | 0.52 |

locations of WBEA stations over the OS region. Evaluation results show biases at various degrees by GM-orig and GM-PRISM simulations. The GM-PRISM method improved surface $SO_2$ predictions relative to GM-orig for the summertime with the correlation coefficient (R) increased from 0.78 to 0.80, and the normalized mean bias (NMB) reduced significantly from -0.42 to -0.06, as summarized in Table 1. GM-PRISM also improved (to a lesser extent) the wintertime surface $SO_2$ predictions relative to GM-orig in terms of mean bias, reducing NMB from -0.19 to -0.16. We note that wintertime results are less conclusive due

to the absence of CEMS emissions and stack parameter data as input for GM-PRISM for the most of the winter period. Model $SO_2$ levels by both GM-orig and GM-PRISM simulations were biased low for most of the WBEA stations. We note that due to strong spatial heterogeneity of concentration fields ($SO_2$), evaluations against observations at individual WBEA stations resulted in diverse statistics. This in turn demonstrates the impact of different plume rise parameterization on modelling the dispersion (transport direction and range) of pollutants. We also note that different choices for the plume parcel convergence

criteria $\rho_{conv}$ result in different levels of performances by the PRISM algorithm. Our tests with PRISM embedded within the GEM-MACH model with $\rho_{conv}$ values of 0.1%, 0.3% and 0.5% resulted in summertime NMB scores of -0.27, -0.06 and 0.17, respectively (see Tables 1, S1 and S2). With $\rho_{conv} = 0.5\%$ resulting in 37% overestimation and $\rho_{conv} = 0.1\%$ resulting in 35% underestimation for surface $SO_2$ concentrations for the full 6-month simulations (including both CEMS and non-CEMS periods), $\rho_{conv} = 0.3\%$ was found to be the optima convergence criteria for our modelling study. $\rho_{conv} = 0.3\%$ resulted in the

relatively small negative bias of 11% over the entire 6-month simulation period, as shown in Table 1.

Several factors may contribute to the overall model negative bias (both with GM-orig and GM-PRISM). These can potentially be related to the performance of the meteorology model in simulating mixing conditions for the same locations and time periods, which would require further investigations, including comparisons to observed surface temperatures and vertical temperature profiles. Another possible reason is the coarse resolution of the model with 2.5 km grid spacing and numerical

dilution of mixing ratios, rendering model generated surface concentrations less representative of near-source observed values. Russell et al. (2019) used GEM-MACH simulations at 2.5 km and 1 km resolutions to demonstrate that increased resolution can result in local increase in concentration, suggesting that model simulations at higher resolutions can potentially improve model performance and reduce the negative bias at the surface. This needs further investigation with simulations at even higher



resolutions (e.g. 50 m, Fathi et al., 2023). A key difference between the summer and winter simulation periods is the availability of time-specific stack parameters from hourly CEMS data (stack parameter, emissions) as input for model simulations, which added further uncertainty for wintertime evaluations. For the summertime period, for which CEMS data were used as input for simulations, the PRISM algorithm improved the predictions significantly both in terms plume final height and surface concentrations, in evaluations against observed values. We note the significance of the improved predictions of plume height by the PRISM algorithm under highly convective and complex summertime conditions (where enhanced turbulence plays a greater role in dispersion relative to the more stable conditions of wintertime).

## 4 Conclusions

In this work, we investigated the behaviour of pollutant plumes emitted from industrial stacks under various atmospheric dispersion conditions in the context of plume rise modelling. As demonstrated in this work, the vertical distribution and downwind dispersion of pollutants emitted from high-temperature anthropogenic sources are controlled by plume parcel buoyancy and water content as well as by ambient atmospheric conditions. We explored the impact of moist thermodynamics on buoyant plume rise from industrial sources, through development of a new plume rise parameterization, PRISM (Plume-Rise-Iterative-Stratified-Moist). This new approach incorporates the thermodynamic effects of latent heat exchange associated with phase transitions of in-plume water in empirical formulations by Briggs (1984), while performing layered (stratified) calculations of parcel buoyancy for the rising plume. The effluents emitted from high-temperature stacks include significant amounts of combustion-generated water vapour that can condense as the plume rises and cools. The subsequent heating due to release of latent heat can prolong the buoyancy of the plumes and result in increased rise above the stack top. Conversely, the evaporation of the entrained liquid water within the plume can result in additional cooling of the effluent and limit the rise. We also note that the addition of condensed water within the plume modifies parcel buoyancy and can act as a rise limiting factor, through latent heat loss as this condensed water evaporates.

As the water emissions data were not available for sources of interest (Canadian oil sands) from the emission inventory datasets, we estimated water emissions from the estimated NOx and $CO_2$ emissions based on aircraft measurements during an aircraft campaign in 2018 over Canadian oil sands (ECCC, 2018). For this purpose we used a stoichiometric ratio of 1:2 of $CO_2$ to $H_2O$, as methane was assumed to be the primary combustion fuel for the considered emission sources. We demonstrated the significant impact of latent heat exchange due to phase changes of in-plume water on plume buoyancy and the final height reached by the pollutant plumes emitted from anthropogenic sources, through standalone (offline) simulations using PRISM with the reported stack source information for several oil sands sources as input data (stack exit temperature, volume flow rate, and estimated water emissions). Our results show that emitted effluents that contain water vapour can rise up to 500 m higher than "dry" (no water content) combustion plumes with the same initial exit momentum and buoyancy (see Fig. 2). We showed that plume behaviour has a stronger dependence on plume parcel water content than effluent exit temperature, suggesting that addition/removal of water mass in both gas and liquid phases can act (and potentially be utilized) as an effective controlling factor for the height reached by anthropogenic pollutant plumes and their downwind dispersion.





We also showed that pollutant plumes can behave differently under dry and humid conditions and in the presence of precipitation, through accounting for the thermodynamic impacts of entrained water (vapour and condensed) from the ambient air into the parcel. Both emitted and entrained water were found to impact plume buoyancy and final rise height, and may boost or
limit the buoyant rise of the plumes. For instance a plume parcel can maintain its water vapour content, and positive buoyancy, for a longer duration and up to higher altitudes under humid atmospheric conditions than dry conditions. Conversely, if water mass (rain droplets, ice, snow) is present in the ambient air, as this water is entrained into the warm plume parcel, it can result in heat loss and latent cooling as the water evaporates (and ice melts) and consequently limit the buoyant rise of the plume. We showed that moist thermodynamics has a wide-ranging impact on plume behaviour and surface $SO_2$ concentrations over a
large region, under varying atmospheric conditions (dry and humid, cold and warm, stable and convective). This was accomplished using a series of retrospective model simulations in which Environment and Climate Change Canada's GEM-MACH air-quality model was used, coupled with the PRISM moist plume rise algorithm (GM-PRISM), for a 6 month period. These modelling results demonstrate the moist thermodynamic impact, with $\pm 100\%$ difference in the average $SO_2$ concentrations near industrial sources (see Figures 4 and S1).

Through comparisons with aircraft observed $SO_2$ plumes during the OSM 2018 airborne campaign we further demonstrated the impact of moist thermodynamics on plume behaviour and showed that accounting for such effects can significantly improve plume height predictions, on average by up to 50% in terms of NMB (normalized mean bias). These impacts were demonstrated to provide a more accurate description of plume rise, through evaluations of model performance against WBEA surface monitoring network data (surface $SO_2$ concentrations) that showed significant improvements for the summertime (and moderate
improvements for the wintertime) simulations in terms of all statistics (e.g., correlation coefficient, bias, see Table 1 and Figure 7). These improvements in predictive capabilities through utilizing PRISM, further reinforces that moist thermodynamics is a key component of the rise of bouyant plumes - and influences the long range transport and surface concentration of emitted pollutants.

For the period between April and July 2018 (inclusive), where hourly (directly measured) CEMS stack parameters and emissions data were available as model input information, the new plume rise algorithm in GM-PRISM simulations outperformed
the older parameterization by 50% in terms of NMB (reduced RMSE by about 50%) in calculating the plume final (equilibrium) height (Fig. 6). GM-PRISM also improved all statistics (R, FAC2, NMB, RMSE – Table 1) for evaluations against the WBEA surface monitoring network data ($SO_2$) for the same period. Evaluations for the wintertime simulations were less conclusive due to the lack of hourly input data (stack parameters, emissions) and aircraft direct observations of the plume heights. The new
plume rise algorithm PRISM is highly sensitive to model input information such as stack parameters and source emission rates. The biases in simulated surface concentrations, especially in the wintertime, may be a function of this missing information. Therefore, further investigation for wintertime conditions, with high-resolution (temporal) and source specific input data are desired as these become available.

This study introduces a novel sub-grid parameterization for plume rise, integrating moist thermodynamics into the iterative
calculation of neutral buoyancy height for plumes emitted from industrial stacks. Our analysis underscores the significant influence of moist thermodynamics on plume rise and the subsequent downwind dispersion of emitted pollutants, thus advancing



our understanding of plume behaviour under different atmospheric dynamics. We also note that the addition of liquid phase water due to condensation can potentially impact the within-plume aqueous phase chemistry and plume composition, which will be further investigated in subsequent research.

*Code and data availability.* The code for the plume rise algorithm PRISM (Plume-Rise-Iterative-Stratified-Moist) used in this work may be obtained on request to Sepehr Fathi (Sepehr.Fathi@ec.gc.ca). The model results are available upon request to Sepehr.Fathi@ec.gc.ca. GEM-MACH, the atmospheric chemistry library for the GEM numerical atmospheric model (©2007–2013, Air Quality Research Division and National Prediction Operations Division, Environment and Climate Change Canada), is a free software which can be redistributed and/or modified under the terms of the GNU Lesser General Public License as published by the Free Software Foundation. The specific GEM-MACH ver-
sion used in this work may be obtained on request to Sepehr.Fathi@ec.gc.ca. The aircraft measurement data from the 2018 campaign used in this work are available from the Environment and Climate Change Canada Data Catalogue (ECCC, 2018). Much of the emissions data used in our model are available online: Executive Summary, Joint Oil Sands Monitoring Program Emissions Inventory report (https://www.canada.ca/en/environment-climate-change/services/science-technology/publications/joint-oil-sands-monitoring-emissions-report.html); and Joint Oil Sands Emissions Inventory Database (https://ec.gc.ca/data_donnees/SSB-OSM_Air/Air/Emissions_inventory_files/) and from ECCC (2023).
More recent updates may be obtained by contacting Junhua Zhang (junhua.zhang@ec.gc.ca).



# Appendix A

**Table A1.** GEM-MACH Model Configuration Details

| Model Component | Description | Reference |
|---|---|---|
| Numerical Weather Prediction model | Global Environmental Multiscale (GEM), v5.1.2 | Côté et al. (1998a,b); Girard et al. (2014) |
| Air-quality model | GEM – Modelling Air-quality and Chemistry (GEM-MACH), based on v3.1.0a2 | Moran et al. (2010); Makar et al. (2015a,b) |
| Model grid and nesting, time stepping | North American 10km res parent domain provides boundary conditions for 2.5km high-res, with 64 vertical levels AB/SK domain. Time-stepping; 10km: 5 min phys., 15 min chem. 2.5km: 1 min phys. and 2 min chem. | Girard et al. (2014); Makar et al. (2015a,b) |
| Weather/Aerosol feedbacks | Direct effect via binary water-dry aerosol mixtures with *Mie* algorithm optical property calculations. Indirect effect via aerosols providing cloud condensation nuclei via the Abdul-Razzak and Ghan scheme. | Abdul-Razzak and Ghan (2002); Gong et al. (2015); Makar et al. (2015a,b) |
| Gas-phase chemistry | Acid Deposition and Oxidant Mechanism, version 2 (ADOM-II) represents gas-phase chemistry for 42 gas species, integrated using a Young and Boris solver. | Stockwell & Lurmann (1989) |
| Particle microphysics | Sectional approach – 8 particle species (sulphate, nitrate, ammonium, primary organic carbon, secondary organic carbon, black carbon, sea salt, crustal material), and 12 particle bins | Gong et al. (2002, 2003) |
| Aqueous chemistry and gas and aerosols scavenging | Cloud scavenging of gases and aerosols, aqueous phase chemistry using a Young and Boris solver (combined time-resolved and steady-state chemistry). | Gong et al. (2015) |
| Deposition | Gas (Robichaud scheme) and particle dry deposition (Zhang scheme) as described in Makar et al. (2018) | Makar et al. (2018) |
| Particle inorganic thermodynamics | Sulphate-nitrate-ammonium non-ideal (high concentration) thermodynamic equilibrium system solved using a nested iterative approach. | Makar et al. (2003) |
| Advection and mass conservation | Chemical transport in GEM-MACH is solved utilizing an implicit semi-Lagrangian (SL) advection space-time integration scheme. The SL scheme is not inherently mass conserving, and therefore requires the use of a post advection mass conservation step (3D ILMC approach used here). | Bermejo and Conde (2002); Sørensen et al. (2013); de Grandpré et al. (2016) |
| Emissions data | Emissions are processed based upon the Sparse Matrix Operator Kernel Emissions – SMOKE; emissions data from hybrid Oil Sands Database. Large stack data derived from Continuous Emissions Monitoring (CEMS). | Coats (1996); Zhang et al. (2018) |
| Plume rise parameterization | Briggs (1984), and PRISM (Plume-Rise-Iterative-Stratified-Moist) as described in this work, were used for calculating plume rise in GEM-MACH simulations. | Briggs (1984) Akingunola et al. (2018) Fathi et al. (2024) |



*Author contributions.* SF: Lead author (coding, scenario simulations, theory development, writing of paper drafts). PM: Theory development, experiment design, review and contributions to paper drafts. WG: Theory development, review and contributions to paper drafts. JZ: Processing of emissions data, provision of stack parameters, advice on interpretation of emissions data. KH: Advice on, collection of and provision of aircraft data. MG: Review and contributions to final paper draft, advice on interpretation of field results for plumes.


*Competing interests.* The authors have no competing interests.

*Acknowledgements.* This work was partially funded under the Oil Sands Monitoring (OSM) Program, sub-project "Integrated Atmospheric Deposition", sub-project A-PD-6-2324. It is independent of any position of the OSM Program.



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
