# Peer review of "The importance of moist thermodynamics on neutral buoyancy height for plumes from anthropogenic sources"

_EGUsphere, 2024_

## Author Comment (AC1)

**EGUSPHERE-2024-1655 | Research article**

Submitted on 01 Jun 2024

**The importance of moist thermodynamics on neutral buoyancy height for plumes from anthropogenic sources**

Sepehr Fathi, Paul Makar, Wanmin Gong, Junhua Zhang, Katherine Hayden, and Mark Gordon

**Handling editor:** Andrea Pozzer, andrea.pozzer@mpic.de

**Author Responses to Referee Comments**

We thank both reviewers for their detailed and constructive feedback. The implementation of the suggested revisions has significantly improved our manuscript. Note that we have also corrected a typo in both the main manuscript and the supplementary document: the coefficient of entertainment for the vertical plume rise is $\alpha = 0.08$. We respond to individual reviewer comments below in detail.

**Color Code**

*RC, Referee Comments in Black*

*AC, Author Comments in Blue*

*EC, Editor Comments in Green*

**RC1: Comments by Anonymous Referee #1 (24 Sep 2024)**

In this paper, the authors present Plume Rise Iterative Stratified Moist (PRISM), an update to the Briggs model which considers the moist thermodynamics of rising plumes. The choice of the Briggs model makes sense because the authors are interested in stack plumes (rather than wildfire smoke plumes for which Freitas would be more suitable). This paper is well written and the introduction clearly communicates why moist thermodynamics is important for stack plumes. In my opinion the paper does a good job comparing PRISM with the previous layered version of Briggs that did not consider water vapor. I was particularly convinced by the confidence ratios. Also, it makes the important point that the PRISM model is sensitive to stack water emissions and that implementing PRISM substantially reduces bias.

This paper starts to lose me when the authors talk about whether GM-orig or GM-PRISM is better at predicting plume top heights, as the authors do not use stack water emissions to drive these models. Lack of data is a totally normal issue, but due to the fact that the PRISM model was so sensitive to stack water emissions, with stack water emissions impacting plume top heights by up to 500m, I am not sure that the bias reduction given by GM-PRISM is as big as the authors report it to be. If stack water is added and GM-PRISM heights go up closer to GM-orig, than the bias reduction might be much smaller.

Therefore, I am recommending the following major revisions.

1. I would recommend that the authors look at observed plume heights for the test cases where they do have water emissions. It looks like for the offline model simulations, stack water emissions were used, could observed plume heights be reported here as well? This would go a long way to convincing me that the bias correction is as large as reported in the current version of the paper.
2. In section 2.1, there seems to be a lot of overlap with the supplement, and the main text and the supplement seem to present the overlapping material in a slightly different order. For instance, line 117-118 says "derivations of the formulae presented here are provided in the supplement" but by this point Equation (1) has been presented, the derivation of which is shown in Supplement S1 Eq1-Eq6. I would recommend making sure that supplement equations are referenced where relevant in the main text, and I would perhaps make it more clear that the supplement is using the discrete forms of the model equations and the main text is using the continuous forms.

3. Throughout the methods and results section, I would suggest that the authors are more clear in differentiating the offline PRISM vs the GM-PRISM. I would make sure the offline and online simulations are mentioned in the intro and section 2.2 and 2.6. It might also make sense to reorganize the methods as follows (using current section numbers): 2.1, 2.2, 2.6, 2.4, 2.3, 2.5.
4. The authors might want to consider a title which indicates that this paper is about developing a new model. That way it could reach the audience of people who might want to apply this model.
5. It might be nice to combine figures 4 and 5, since they have a lot of the same information. I liked the format of Fig 5 and the error bars in Fig 4.

Overall, nice work and I look forward to seeing a revised version.

**AC1: Author Responses to RC1 (December 12, 2024)**

We thank the reviewer for their comments and the overall "nice work" – greatly appreciated. We also appreciate the detailed and constructive feedback. The recognition of our PRISM model's value and the positive comments on our paper's clarity and structure are highly valued. We have implemented the suggested revisions, which have strengthened our work.

**AC1.1**

During the review process, during the period while we were awaiting comments from reviewers, we obtained additional data and developed a more complete input dataset that included estimated water emissions and hourly CEMS stack parameters for the stacks in the 2.5km GEM-MACH domain. These new data were used in GEM-MACH simulations (these data were not available at the time of initial submission of the manuscript; we were anticipating the reviewers' request). This brought the GEM-MACH simulations to the same degree of capture of water data as was done for standalone simulations for the original submitted preprint manuscript. We performed GEM-MACH model simulations for the study period using the new/revised input dataset with both plume rise parameterization options (GM-orig and GM-PRISM) and reported the results in the revised version of the manuscript, modifying the original table of model statistics to include the results with the use of water emissions for the additional GEM-MACH stacks. The inclusion of the CEMS parameters and estimated combustion water emissions, as presented in the revised manuscript, confirms the improved performance of the new PRISM method. In fact, evaluations against WBEA monitoring network data of $SO_2$ concentrations show even better performance by GM-PRISM when driven by CEMS hourly parameters and estimated water emissions. The revised Table 1 in the revised manuscript contains the revised estimates, using the new data: the track changes version of the revised manuscript shows the changes in performance associated with the new data. Note that the GM-Orig values also changed – these were due to the CEMS data updates, since GM-Orig does not make use of the water emissions to calculate plume rise.

**AC1.2**

We thank the reviewer for the good suggestion of additional referencing to the corresponding SI discrete formulations in the methods section. The following revisions were made to the main text following the reviewer's suggestion:

*Text added to the paragraph after Eq. (1)*
(see supplement S1, Eq.1 to Eq.6 for the derivations and the corresponding discrete formulations)

*Text added to the paragraph after Eq. (2)*
(see S1, Eq. 7 for the corresponding discrete formulation)

*Text added to the paragraph after Eq. (3)*
(derivations of the formulae presented here are provided in the Supplement section S1, see Eq. 8 to Eq. 19)

**AC1.3**

Following the suggestion by the reviewer, we have reordered the subsections in the Methods section in the following order based on original numbering: 2.1, 2.2, 2.6, 2.4, 2.3, 2.5. Note that we have also combined the previous sections 2.3 and 2.5 into the revised manuscript's 2.5 section "Aircraft Campaign and WBEA Surface Monitoring Network". In addition, we added more text throughout the main manuscript to further distinguish between standalone PRISM and GM-PRISM simulations (see for example lines 235 to 274, which now refer to results solely from GEM-MACH (GM-PRISM and GM-orig) simulations in the "Track changes" manuscript version. We have also broken up the manuscript so that the previous section 3.1 has been subdivided into stand-alone PRISM tests (revised manuscript section 3.1, **Model Sensitivity to Plume Rise Parameterization: standalone PRISM simulations**) and GM-MACH tests (revised manuscript section 3.2, **Model Sensitivity to Plume Rise Parameterization: GEM-MACH simulations**).

**AC1.4**

The reason for the choice of the title in the manuscript reviewed followed some discussion with the handling editor on our making the case that the manuscript was more suitable for ACP rather than GMD. We have pointed out for example in the submitted manuscript that we have identified a physical process in plume rise which had to date been unrecognized – and hence the work has broader interest than to the modelling community alone. As part of this discussion and in response to a request from the handling editor on 12 Aug 2024 (see below), we had revised the original manuscript title "**The influence of moist thermodynamics on neutral buoyancy height for plumes from anthropogenic sources: investigation using PRISM (Plume-Rise-Iterative-Stratified-Moist)**" to "**The importance of moist thermodynamics on neutral buoyancy height for plumes from anthropogenic sources**". Consequently, we feel we should leave the title of the manuscript in the second form as submitted, following the Editor's recommendation.

However, we will make sure to include relevant keywords during the final submission of the manuscript related to the model development aspect of this work, to reach that part of the audience that are specifically interested in our modeling approach, and we are making the PRISM code available by email request for interested parties. We thank the reviewer for raising this point.

"

**EC**

**12 Aug 2024**
**Editor initial decision: Start review and discussion after technical corrections**
**by Andrea Pozzer**
**Justification (visible to authors and reviewers only):**

**Dear author,**
**following the changes made to the manuscript, I accept it for discussion in the EGUsphere. As additional minor change, I would like to ask you to change the title of the manuscript and make it more general. Following the "Guidelines for authors" (https://www.atmospheric-chemistry-and-physics.net/policies/guidelines_for_authors.html), the title "[...] should be concise and**

consistent with the content and purpose of the article. For research articles, ACP prefers titles that highlight the scientific results/findings or implications of the study." Therefore I suggest the following titles, similar to the original one: "Moist thermodynamics is important for the neutral buoyancy height for plumes from anthropogenic source" or "The importance of moist thermodynamics on neutral buoyancy height for plumes from anthropogenic source" the latter being more neutral.

Best regards"

**AC1.5**

In response to the suggestions by both reviewers, we have combined Figures 5 and 6 into the following new Figure 5

[Figure]

**Figure 5. Predictions of plume height in GM-orig and GM-PRISM simulations compared to OSM 2018 aircraft observations for the eleven case studies. For each hour of the flight, aircraft observed vertical profiles of concentration are shown as density maps (white to blue) up to the height visited by the aircraft by that hour. Concentrations are shown as shaded blue regions, which have been normalized to the maximum concentration encountered during the flight. Aircraft observed SO₂ plume maximum concentrations heights are marked with cyan stars, and are taken here to represent the observed plume heights. Plume maximum concentration height predictions by GM-orig (grey circles) and GM-PRISM (orange squares) are compared with**

**RC2: Comments by Anonymous Referee #2 (23 Oct 2024)**

This study introduces a new approach (PRISM - Plume-Rise-Iterative-Stratified-Moist) for modeling how industrial smoke plumes rise and spread in the atmosphere, with a special focus on how water content affects plume behavior. The water vapor condensation and evaporation were often overlooked in previous models but shown to be crucial for plume rise dynamics. This study makes a valuable contribution to air quality modeling by demonstrating the importance of including water vapor effects in plume rise calculations, with scientific ground evidence from both theoretical analysis and observational comparisons. Overall, the paper is well written, and the model experiments are well structured. I recommend the publication of this paper and have a few minor suggestions.

1. The authors have mentioned about two different plume rises ("vertical" and "bent-over") in section 3.1 and different formulations for these two types in section 2.1. Maybe it is obvious for the authors, but I think it worths a brief explanation of their differences to benefit broader audience.
2. The choice of density convergence criteria (rconv) may be critical in determining the final plume height. Is it scientifically rigorous enough to conclude that including water content improves plume rise predictions when this conclusion is based on simulations using $\rho conv = 0.3\%$, a value chosen because it produced better matches with observations? Based on the sensitivity studies on different values of rconv shown in Table S1 and S2, it is difficult to conclude that GM-PRISM performs better than GM-orig. I suggest the authors to provide more scientific justification for the rconv = 0.3% selection.
3. I assume Fig. 5 and 6 are showing overlapping information, is it correct? Why not combine these two figures and show them at the same time and altitude scales like Fig. 6 but with the error bars like Fig. 5?
4. It is very difficult to compare the bias and see the station IDs in Fig. 7b and 7d. Is it possible to show in other formats such as bar chart?

**AC2: Author Responses to RC2 (December 12, 2024)**

We thank the reviewer for their encouraging feedback on our study. We are pleased that the reviewer found our PRISM approach valuable for modeling plume dynamics and appreciated the inclusion of water vapor effects. The reviewer's positive comments on the structure and writing of our paper are greatly appreciated. We also value the suggestion of minor revisions, which we have incorporated to enhance our work. We appreciate the recommendation for publication following these minor revisions.

**AC2.1**

Additional text describing the two types of plume rise was added to section 2.1, the paragraph before Eq. (2):

"**Vertical plumes occur when the buoyancy and momentum of the emitted gases are strong enough (and/or the wind speeds are sufficiently low) to overcome the effects of wind. This typically happens under stable atmospheric conditions or when the stack emissions are**

significantly hotter and faster than the surrounding air. The plume rises vertically, under these conditions, until it reaches the neutral buoyancy height, where the plume parcel density approaches the ambient air density. Bent-over plumes, on the other hand, occur when the wind speed is strong enough to bend the plume horizontally. This is more common under neutral or unstable atmospheric conditions. The plume initially rises due to its buoyancy and momentum but is then bent over by the wind, creating a trajectory that is more horizontal than vertical."

**AC2.2**

The following text was added to section 3.1 to further elaborate on this:

"In our standalone tests with PRISM, we have noticed the asymptotic offset between parcel air density and the ambient air density, which depends on the vertical resolution at which buoyancy reduction calculations are performed, falls between 0.1% to 0.5% of ambient air density. That is to say, when parcel density starts to asymptotically approach the ambient air density, as a result of the finite resolution of the calculations and the slight excess humidity within the plume parcel, plume density remains offset from the ambient density within a fraction of a percent of the ambient air density at those heights, although following the same lapse rate as the ambient air. Our criteria for convergence is thus based on the observed numerical behaviour of the rising parcel. We believe that the physical reason for observed situation where the parcel comes to rest without asymptotic rise may reflect detrainment of parcel water to the ambient atmosphere. Future work will focus on evaluating the detrainment impact."

**AC2.3**

In response to the suggestions by both reviewers, we have combined Figures 5 and 6 into the following new Figure 5

[Figure]

**Figure 5. Predictions of plume height in GM-orig and GM-PRISM simulations compared to OSM 2018 aircraft observations for the eleven case studies. For each hour of the flight, aircraft observed vertical profiles of concentration are shown as density maps (white to blue) up to the height visited by the aircraft by that hour. Concentrations are shown as shaded blue regions, which have been normalized to the maximum concentration encountered during the flight. Aircraft observed SO₂ plume maximum concentrations heights are marked with cyan stars, and are taken here to represent the observed plume heights. Plume maximum concentration height predictions by GM-orig (grey circles) and GM-PRISM (orange squares) are compared with aircraft observed heights (flight median). Results are shown as normalized mean bias (nmb) and normalized root mean squared error (nrmse).**

**AC2.4**

In response to reviewer's comment, font and marker sizes were increased for station IDs in Fig. 7b and 7d.